# Regression with Highly Correlated Predictors: Variable Omission Is Not the Solution

**DOI:** 10.3390/ijerph18084259

**Published:** 2021-04-17

**Authors:** Mariella Gregorich, Susanne Strohmaier, Daniela Dunkler, Georg Heinze

**Affiliations:** 1Section for Clinical Biometrics, Center for Medical Statistics, Informatics and Intelligent Systems, Medical University of Vienna, 1090 Vienna, Austria; mariella.gregorich@meduniwien.ac.at (M.G.); susanne.strohmaier@meduniwien.ac.at (S.S.); daniela.dunkler@meduniwien.ac.at (D.D.); 2Center for Public Health, Department of Epidemiology, Medical University of Vienna, 1090 Vienna, Austria

**Keywords:** correlated predictors, exposure-response association, collinearity, nonlinear effects, multivariable modelling

## Abstract

Regression models have been in use for decades to explore and quantify the association between a dependent response and several independent variables in environmental sciences, epidemiology and public health. However, researchers often encounter situations in which some independent variables exhibit high bivariate correlation, or may even be collinear. Improper statistical handling of this situation will most certainly generate models of little or no practical use and misleading interpretations. By means of two example studies, we demonstrate how diagnostic tools for collinearity or near-collinearity may fail in guiding the analyst. Instead, the most appropriate way of handling collinearity should be driven by the research question at hand and, in particular, by the distinction between predictive or explanatory aims.

## 1. Introduction

Correlation between independent variables in multiple regression modelling can have a far-reaching impact on the accurate estimation of the model and, thus, its results and interpretation [1]. This often neglected phenomenon has been recognised as ‘collinearity’, but has rarely been addressed in data analyses with the adequate attention it deserves. Strictly speaking, collinearity or ‘ill-conditioning’ (as termed by Draper and Smith [2]) refers to linear dependence between two independent variables. The notion can be extended to ‘multicollinearity’, meaning that multiple independent variables are involved. The casual use of ‘collinearity’ to denote a situation of strongly but not perfectly correlated variables has led to a distinction between exact and near-collinearity, of which latter is indicated by strong but not perfect correlation between a pair of independent variables [2]. Exact collinearity leads to non-unique least-squares estimates of regression coefficients while near-collinearity can cause numerical problems when fitting the model. Under the latter condition, the least-squares estimates can have a substantially inflated variance [3], and with increasing degree of collinearity, the least-squares solution becomes more and more unstable, and regression coefficients more and more sensitive to small changes in the data. In such a case, the interpretation of a regression coefficient in a multivariable model, which is based on varying one variable while keeping the others constant, becomes more difficult since variations in one variable are associated with dramatic shifts in another variable. Small data samples are especially sensitive to collinearity as inflation of standard errors of regression coefficients induced by collinearity adds up to a generally more unstable estimation.

Exact collinearity may occur if variables are naturally related to each other. For example, in blood analysis, white blood cells consist of five subtypes: neutrophils, eosinophils, basophils, lymphocytes, and monocytes. If the concentrations of all five subtypes and of total white blood cells are measured and included as independent variables in a regression model to predict C-reactive protein (CRP), a software package will ‘automatically’ detect exact collinearity by linear programming and then arbitrarily choose one variable, often simply the last one, to be omitted from the model (see Table 1 and Appendix A). For near-collinearity, however, recommendations include the use of a summary variable composed of several correlated independent variables in a regression model, or the omission of one or several variables assuming the dropped variable does not hold further valuable information. Hence, while collinearity has been studied in many articles in the past (e.g., [4,5,6,7,8]), treatment of the problem in practice mostly targets the prevention of symptoms (e.g., inflated variance, unstable parameter estimates). We propose a more purposeful approach to dealing with collinearity in statistical modelling by distinguishing description from prediction or explanation [9,10].

Therefore, the objective of this article is to revisit collinearity diagnostics and associated recommendations to deal with collinearity by focusing on two simple case studies each with two independent variables, but with different types of research aims: diagnostic prediction and explanation. In particular, we will evaluate the use of the variance inflation factor (VIF), the condition number and a more refined method to detect high, possibly nonlinear correlation between independent variables using a generalised additive model. The first illustrative example uses data from a study by Tang et al. [12] who have aimed at identifying quantitative features derived from computer tomography (CT) chest images of 176 confirmed COVID-19 cases in order to perform binary classification of disease severity within this cohort. The second example is focused on the current anthropogenic contribution to the progression of climate change.

The remainder of this work is organised as follows. In Section 2, we will review the problem of collinearity in the context of model-building including well-known diagnostics and remedies and will use the blood analysis example for illustration. The two case studies will be presented in Section 3 to demonstrate how collinearity diagnostics would guide further analyses. Some concluding remarks are given in Section 4.

## 2. Methods

Previous discussions of ill-conditioned data and their consequences have focused on three key aspects: impact of collinearity on statistical estimation and inference, diagnostics to detect it, and proposals to resolve it [2,4,5,7,13,14]. The following section will adhere to this general structure with the intent to give insight into the basic considerations when dealing with highly correlated variables in regression analysis.

### 2.1. The Problem of Collinearity

We will briefly describe exact and near-collinearity. Consider the classical multiple linear regression model with p independent variables and sample size n given in matrix notation by
(1)y=Xβ+ϵ
where $y\in R^{n}$ denotes the dependent variable, $\epsilon \sim N\left(0,\sigma^{2}I\right)$ refers to the error term with the n×n  identity matrix I, and $X\in R^{n\times \left(p+1\right)}$ denotes the combination of the n×p  design matrix of the independent variables ${\left(x_{1},\ldots ,\;x_{p}\right)}^{T}$ and a unit vector $x_{o}={\left(1,\ldots ,1\right)}^{T}$ in the first column. The vector $\beta ={\left(\beta_{0},\ldots ,\;\beta_{p}\right)}^{T}$ represents the p+1 regression parameters of the model.

The standard estimation approach to minimise the residual sum of squares is the well-known ordinary least-squares (OLS) estimation process aiming to minimise the residual sum of squares (RSS)
(2)$$RSS\left(\beta \right)={\left(y-X\beta \right)}^{T}\left(y-X\beta \right).$$

The closed-form expression of the estimate for the true unknown regression coefficients can then be obtained by the first derivation of Equation (2).
(3)$$\hat{\beta }={\left(X^{T}X\right)}^{-1}X^{T}y$$

Suppose one variable of the design matrix X  is exactly equal to a linear combination of the remaining variables, e.g., white blood cell count exactly equals the sum of concentrations of granulocytes (neutrophils, eosinophils, basophils), lymphocytes, and monocytes (Table 1). Then, $X^{T}X$ is singular, i.e., has rank less than the number of its columns and cannot be inverted. Hence, $\hat{\beta }$ cannot be computed unless one of the variables in X is omitted from the analysis.

The variance–covariance matrix of the OLS estimator can be written as:(4)$$Var\left(\hat{\beta }\right)={\left(X^{T}X\right)}^{-1}E\left[e^{T}e\right]{\left(X^{T}X\right)}^{-1}$$
and reduces to
(5)$$Var\left(\hat{\beta }\right)=\sigma^{2}{\left(X^{T}X\right)}^{-1}$$
under the assumptions of homoscedasticity and no autocorrelation among the OLS residuals $\left(E\left[e^{T}e\right]=\sigma^{2}I\right)$. In the case of linear dependence among the columns in the design matrix  X, it is obvious that both Equations (4) and (5) cannot be numerically evaluated as $X^{T}X$ cannot be inverted. Now suppose we just consider white blood cells and neutrophils for the analysis. These two variables still have a correlation coefficient of 0.982 and $X^{T}X\;$ is close to singularity. The term is invertible, but the variance of the estimated regression coefficients of white blood cells and neutrophils will be outstandingly high. Hence, near-collinearity is sufficient to cause inflated standard errors and wide confidence intervals which, in turn, reflect that slight changes in the data can cause considerable variations in the parameter estimates [15]. However, near-collinearity is mainly defined as a matter of the degree of interrelation among the columns in the design matrix. Therefore, the question arises at which threshold of bivariate correlation should remedies be considered.

### 2.2. Diagnostics for Collinearity

Identifying the degree of interrelations between the independent variables is an important part of data screening prior to analysis [16]. Traditional diagnostics comprise checking the pairwise sample correlation matrix and investigating the variance inflation factor (VIF). As a pragmatic condition for ruling out near-collinearity, all pairwise correlations between independent variables intended to be included in the model should be clearly smaller than 1. According to Dohoo et al. [17], values above 0.9 almost certainly point towards the presence of collinearity. However, the proposed requirements present sufficient conditions, not necessary ones, for the presence of collinearity. In the blood analysis example, all pairwise correlations are smaller than 1, but still exact collinearity exists. Similarly, pairwise correlation coefficients can only partly detect near-collinearity.

A stricter criterion is obtained by evaluating the multiple correlation between covariates, i.e., the squared coefficient of determination $R_{i}^{2}$ of each model where one independent variable, $x_{i}$, is regressed on all other independent variables $M_{i}=\left\{x_{1},\;\ldots ,x_{i-1},x_{i+1},\ldots ,x_{p}\right\}$. These numbers can be transformed into variance inflation factors [18] which express the magnitude by which variances of regression coefficients inflate when independent variables are correlated:(6)$$VIF\left(\beta_{i}\right)=\frac{1}{1-R_{i}^{2}}\;.$$

If the independent variables in $M_{i}$ are uncorrelated to $x_{i}$, we will obtain $R_{i}^{2}=0$ and, hence, $VIF\left(\beta_{i}\right)=1$. In the case of exact collinearity (as in the blood analysis example), $R_{i}^{2}=1\;$ and $VIF\left(\beta_{i}\right)=\infty$. A rule-of-thumb claims that VIF values should all be lower than 10 (corresponding to $R_{i}^{2}<0.9$), while values between 5 and 10 are already considered problematic [14,19]. However, O’Brien [15] argues that model specification should not end with proving that the rule is met, and other issues with model building may also impact the stability of the estimates. In the reduced model with only neutrophils and white blood cells, VIF for both variables results as $1/\left(1-0.982^{2}\right)=28.02$ indicating problems with near-multicollinearity according to this rule.

An alternative diagnostic proposed by Belsley et al. [1] is the condition number defined as the square root of the ratio of the largest ($\lambda_{max}$) to the smallest ($\lambda_{min}$) eigenvalue of the scaled design matrix. A condition number of around 10 gives evidence to weak dependencies, a value in between 30 and 100 indicates moderate to strong interdependencies, and a value above 100 points towards the presence of collinearity [20]. Condition indices and the so-called ‘regression coefficient variance-decomposition proportions’ are extensions of this metric that provide more insight into the variables responsible for collinearity in multivariable situations with more than two variables. The interested reader is referred to [1] for further details. The condition number in the blood cell example was 23.28, confirming the conclusion from investigating the VIF.

Additional potential indicators of collinearity less often mentioned in the literature are obtained by checks of deviations from expected model behaviour against prior subject-matter knowledge (e.g., unexpected signs of coefficients) or by observing a large change of effect size caused by little alterations in the data matrix [2,14].

In addition, similarity checks of shared information considering non-monotone relationships among the independent variables can identify unnoticed patterns of association. One such approach is redundancy analysis. In this approach, also, variable-specific $R_{i}^{2}$ are estimated to investigate how well one variable can be predicted from the others, but instead of linear regression, generalised additive regression models [21] are used where continuous variables are modelled flexibly using restricted cubic splines instead of just assuming a linear relation between variables. The most predictable variable is then omitted in a stepwise manner. The approach is described in more detail in Harrell ([13], p. 80) and has been implemented in the R-function *redun* of the package *Hmisc* [22].

### 2.3. Remedial Measures for Collinearity

Recommended solutions in the presence of highly correlated independent variables involve (a) the omission of one of the affected variables from the analysis, (b) combining the strongly correlated variables into a single composite score or (c) switching to more adequate modelling approaches able to handle correlated variables such as principal component analysis (PCA), ridge regression or partial least-squares (PLS) regression [2,4,7,14]. A widely recommended and applied solution to near-collinearity is the omission of a “redundant” independent variable (data reduction). Tabachnick et al. [23] state that after deletion of one of the two redundant variables, the problem of near and exact collinearity can be considered solved when dealing with strong bivariate correlation such as 0.9 and higher. However, the authors suggest the omission of one variable or the creation of a composite score already for bivariate correlation around 0.7. The simplicity of variable pruning is alluring to researchers seeking a quick and easy remedy for the issue at hand because quite advanced approaches, such as PCA or ridge regression, may go beyond the typical level of statistical knowledge of health researchers. Alternatively, an adequate transformation of the correlated variables into a single index is not always feasible, and the latter approaches result in a loss of interpretability of the individual contribution of the variables to the dependent variable [4,7].

In the blood analysis example, a simple solution that uses subject matter knowledge is to reparametrise the model such that instead of neutrophils and total white blood cell counts, neutrophils and the difference of white blood cell count and neutrophils are used as independent variables. These two variables exhibit a correlation of 0.668 and a VIF of only 1.806. In this way, the information contained in the two variables is fully retained, and the interpretation of the regression coefficients of the new components is even more appealing as before: for example, the regression coefficient of neutrophils now corresponds to the expected difference in CRP corresponding to a difference of 1 G/L of neutrophils, given equal concentration of other components of white blood cell count. The analysis of this data set can be found in the Appendix A.

## 3. Examples

### 3.1. Worked Example: COVID-19 Study

In the study by Tang et al. [12], a set of potentially disease-associated features was extracted from 134 chest computer tomography (CT) images of 176 confirmed COVID-19 cases using machine learning techniques. These cases were classified as severe (N = 55) or non-severe (N = 121) based on the clinical course of the disease. Two CT-derived features were considered of higher importance: the volume of ground-glass opacity (GGO) regions and the consolidation region defined by a specific Hounsfield unit (HU) spectrum of the CT densities (GGO region: HU [−750,−300]; consolidation region: HU[−300,50]). Suppose we are interested in predicting severity of the disease with these two continuous variables using a logistic regression model. The joint distribution of the two relevant independent variables is depicted in Figure 1. Both variables exhibit right-skewed distributions (and hence, Figure 1 shows their square roots).

The Pearson correlation coefficient between GGO and consolidation region is 0.76, suggesting a strong linear relationship between the two variables. Hence, the VIF for both variables is 1.75, indicating no issue with near or exact collinearity. Likewise, redundancy analysis using generalised additive models results in R^2^ of 0.618 and 0.600 if GGO or consolidation are predicted by the respective other variable. Similarly, the condition number is 4. Hence, none of the three criteria would suggest omitting any of the two variables in the model.

Table 2 contrasts the multivariable model with the two univariable models. The regression coefficient of the variable consolidation changes by −102% and is no longer significantly associated with the dependent variable disease severity when adjusted for GGO in the multivariable model. Such a deletion of effect size could be another indicator of multicollinearity as mentioned in Section 2.2. The model with GGO region alone achieves a lower value of the Akaike information criterion (AIC) than the model with both variables, and this comparison suggests omitting the consolidation region, contrary to the suggestion of the collinearity detection criteria. Similarly, the C-index of the model with only GGO region is not lower than that of the model with both variables. This example illustrated that for models aiming at providing accurate predictions, information criteria such as AIC may be useful quantities to guide model building, but collinearity checks are not sufficient to make decisions on variable selection.

### 3.2. Worked Example: Carbon Emissions and Temperature Anomalies

Ill-conditioning in model building is often further complicated by nonlinear effects or autocorrelation, both of which we will illustrate in our second illustrative example. Research suggests that global temperature has been subject to long-term fluctuations in the past already before the evolution of mankind but has recently been heavily influenced by increasing carbon dioxide emissions (CO_2_). Here, we define a hypothetical study in which we want to disentangle the two effects by means of analysis of time series of annual global temperature anomalies and annual global CO_2_ emissions. In particular, we would determine to which extent temperature anomalies can be explained by CO_2_ emissions, and in the analysis, we will use temperature anomalies as the dependent variable and globally emitted CO_2_ as the main independent variable. To separate the effect of emissions from long-term fluctuations, we will adjust for calendar year. Calendar year itself probably indirectly causes two anthropogenic drivers of CO_2_ emissions: advances in technology and growth in world population size. If we assume that there are no other common causes of temperature anomalies and carbon emissions than calendar year (exchangeability assumption) and if, theoretically, any amount of CO_2_ emission could have been observed in any calendar year (positivity and non-interference assumptions [24]), then the effect of emissions on temperature could even be interpreted as causal effect. However, these assumptions are probably not realistic. Hence, it is important to emphasise that this illustrative data example is not intended as a reliable source to disentangle the effects of anthropogenic and natural contributions to climate change but rather serves to demonstrate problems related to (near-) collinearity. It goes well beyond the scope of this paper to reveal the true relationship between released CO_2_ over the course of the last century and the currently observed climatic changes. We refer the interested reader to Manabe and Wetherald [25] and the special report of the Intergovernmental Panel on Climate Change (IPCC) on the impacts of global warming of 1.5 celsius degrees above pre-industrial levels and related global greenhouse gas emission [26].

Our illustrative analysis is based on publicly available data on climate change distributed with the dslabs R package (data set ‘temp carbon’). The data set contains the annual relative temperature anomalies in centigrade from 1880–2018 over land, ocean, and globally with respect to the 20th century mean temperature [27] as well as the annual cumulative emissions estimate of CO_2_ released to the atmosphere from fossil fuels and cement production from 1751–2014 provided by Boden et al. [28]. In our analysis, global temperature anomalies measured per year will be the dependent variable, and annual CO_2_ emissions given in millions of metric tons and calendar year of measurement constitute the independent variables. In total, the data set contains 268 observations, but only 135 with both carbon emissions and global temperature anomalies (calendar years 1880–2014, Figure 2), which we will be the basis of our analysis.

In Figure 2, the slopes of both time series reveal a sharply increasing trend from the 1950s onwards. In contrast to the fairly smooth time series of carbon emissions, temperature anomalies exhibit severe annual fluctuations. The correlation coefficient between carbon emissions and year is 0.93 and indicates a strong linear relationship. The corresponding VIF is 7.071. As stated before, according to Dohoo et al. [17] and Tabachnick et al. [23], the high correlation between the two independent variables very likely indicates the presence of collinearity and suggests omission of one variable of the pair. Redundancy analysis using generalised additive models suggests the removal of the variable carbon emissions due to a higher adjusted R^2^ value of 0.988. The estimated R^2^ of the generalised additive model regressing year on modelled carbon emission was 0.954. Further, the condition number for this example is 303.3, indicating near-dependency.

Since the research aim was to explain temperature anomalies by changes in annual CO_2_ emissions while adjusting for time itself, none of the two univariable models considering either calendar year or CO_2_ emissions alone is appropriate, despite the recommendation of the collinearity detectors to remove one variable.

Equation (5) follows from Equation (4) only under the assumptions of homoscedasticity and no autocorrelation of the error terms. If these prerequisites do not hold, inference based on Equation (5) will be biased. As the data stems from time series, there was evidence for first-order autocorrelation (Durbin–Watson test: t = 0.71, *p* < 0.001). Hence, we used a quadratic spectral kernel-based heteroscedasticity and autocorrelation consistent (HAC) estimator as implemented in the R package sandwich [29,30] to compute the empirical covariance matrix for the parameters of each model. The robust HAC estimator ensures consistent estimates of the covariance of the model parameters year and CO_2_ emission despite the slight deviations from the underlying model assumptions.

Natural cubic splines with varying degrees of freedom were used to model the nonlinear relationship between each of independent variables and the dependent variable, and tested against a model assuming linear association using the AIC. The AIC criterion suggested to include year without transformation and to use natural cubic splines with 5 degrees of freedom for annual CO_2_ emissions.

Figure 3 displays the partial effects of the two independent variables on temperature anomalies [31]. The year 1975 was when the global temperature was closest to the 20th century mean. Therefore, to describe the effect of year, we fixed global CO_2_ emission of 4596 mt (the emission of 1975) [32]. Likewise, we fixed the year at 1975 to illustrate the adjusted effect of CO_2_ emission on temperature anomalies. The model suggests that temperature increases by 0.009 (95% CI: (0.003, 0.015); *p*-value = 0.006) centigrade each year irrespective of the emitted CO_2_. Furthermore, temperature anomalies are related to CO_2_ emissions in a nonlinear way. While temperature is negatively associated when emissions increased from 0 to around 4000 mt, emissions above 4000 mt appear to be positively associated with temperature, reaching a plateau at about 6500 mt. However, sparse data support for such high CO_2_ emissions led to a wide confidence band which precludes any clear conclusions in that area. Similarly, the confidence bands are also wide at the lower extremes both for year and CO_2_ emissions, but can be explained by the low variability in temperature anomalies of earlier years compared to years after the 1920s (see Figure 3). Our analysis assumes that there are no further common causes of CO_2_ emissions and temperature anomalies (e.g., other types of emissions caused by the constantly growing world population). Under this assumption, the described effect of year is conceptionally interpreted as a controlled direct effect. However, the causal relation between these variables is certainly more complex. Therefore, we once again emphasise the illustrative purpose of this analysis: if a statistical model is fitted to disentangle different causes of a dependent variable, then even near-collinearity between the independent variables does not justify variable omission.

## 4. Discussion

In this article, we have demonstrated, using two examples, how the diagnostic tools for collinearity and near-collinearity may fail in guiding the analyst in selecting the most appropriate way of handling collinearity. In particular, the analysis of the examples revealed that depending on the situation, misguidance may lead to keeping a clearly redundant variable in a prediction model, or to omitting a variable that would be needed for proper adjustment in an explanatory model. Hence, it is the aim of the analysis that should guide decisions on keeping or omitting a variable in a model.

When statistical modelling is used to pursue a predictive aim, two highly correlated independent variables will lead to high variance in the predictions, even if both variables are relevant for prediction. In small samples, it may then be beneficial to omit one of the pair in order to decrease that variance, even if this incurs some new bias in the predictions. This variance–bias trade-off is often expressed as mean squared error (MSE), which is defined as the expected squared deviation of an estimate from its estimand. It can also be expressed as $\;\mathrm{MSE}\;=bias^{2}+\mathrm{variance}$ [33]. Several factors are relevant for the trade-off, the most important being the correlation between the two variables in question, the strength of the association of the variables with the dependent variable, the sample size, and the relative impact of noise (residual error). In the COVID-19 study, consolidation region was hardly associated with the dependent variable once adjusted for GGO region. Hence, despite its marginal association revealed by univariable analysis, the model fit of the prediction model was not affected by its omission, and the AIC even indicated a better fit. Nevertheless, one should not draw overly general conclusions from this result.

In explanatory models, one is usually interested in estimating one of the regression coefficients with high precision or, more generally, to estimate what would happen to the dependent variable if one changes the independent variable of main interest (the ‘exposure’) while other variables (the ‘confounders’) are held constant. Variable selection for such models should be based on the classical assumptions of causal inference, such as conditional exchangeability and positivity [24,34]. For our example, we have drastically simplified possible causes of climate change (apart from CO_2_ emissions) into an assumed association of calendar time with temperature anomalies. Such an association is plausible given that there is evidence for temperature variation long before human beings appeared in the history of the earth or, as put by Sun and Bryan [35], ‘if anything has been constant with regard to the state of the climate system, it is that is has always been changing’. Hence, we adjusted for calendar time to be able to disentangle anthropogenic from other effects on temperature anomalies given our assumptions. In this scenario, by removing calendar time from the model, it would not be possible to answer the study question.

Shmueli [10] and Hernan et al. [9] also mention a third purpose of modelling, descriptive modelling. Here, a researcher is interested in providing a parsimonious description of the main associations in the data sets, and the selection of variables could be more data-driven than in explanatory modelling, where it is exclusively based on causal assumptions. Recommendations for variable omission in case of near-collinearity may stem from that descriptive point of view, but we agree with O’Brien [36] that variable omission in the case of collinearity should not be seen as the default solution. We summarised options to handle collinearity with respect to the research aim in Table 3.

Increased standard errors due to the inclusion of correlated independent variables in the regression model sometimes have to be accepted in order to adequately answer a research question. Excluding relevant variables will also lead to biased and inconsistent parameter estimates if the analysis requires all variables to be included in the model. The quick fix of variable omission provides an easy solution, but can lead to not answering the actual research question at hand or be intentionally misleading by dropping variable(s) such that the results of the presented final model support the researcher’s preferred hypothesis. Therefore, variable omission should not be the default solution for strong correlation between independent variables. O’Brien suggested to assess the “model influence” criterion [36] by checking a few alternative models for deviations from the original model with regards to sign and magnitude of the effect sizes of main interest and shifts in major conclusions. This could be extended to predictions from the model. Substantial changes would suggest keeping the dropped variable. If the set of variables to be included follows from causal assumptions, such competing models may still differ in the way continuous variables are parametrised (more or less flexible), or which interactions between variables are considered. Even in the presence of near-collinearity, multiple regression can still be carried out, and ambiguity regarding the interpretation of the regression coefficients can be noted. However, this necessitates transparent reporting of the existing correlation structure and its impact on the regression results should be stated as the analysis proceeds. Overall, variable omission as a consequence of initial data analysis [16] should be avoided when such omission would affect the ability to answer the research question of interest.

In some examples, as in the blood analysis study, it is possible to remove near-collinearity by reparametrising the independent variables. This reparametrisation can often be guided by domain expertise and, such as in our example, even lead to more attractive interpretability of the regression coefficients. If reparametrisation is not an option, ideal but often unattainable remedies of collinearity would consist of revising the study design or repeating or extending the study such that dependency of variables is largely avoided or that higher accuracy of the parameter estimates is obtained by collecting a larger sample. However, these two suggestions are often not a viable option, and the course of action should then be driven by the research question and not the complexity of the remedy.

## Figures and Tables

**Figure 1 ijerph-18-04259-f001:**
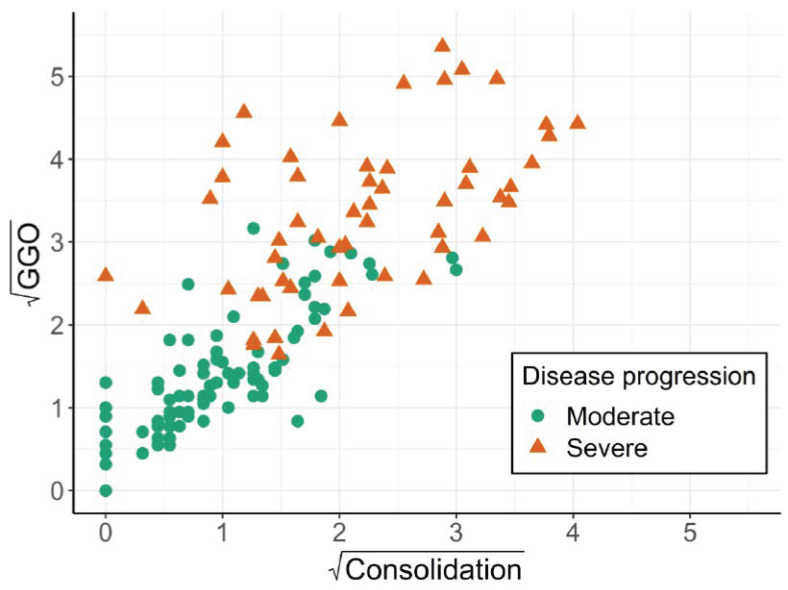
COVID-19 study: scatterplot of the square roots of GGO and consolidation by severity of COVID-19 disease progression.

**Figure 2 ijerph-18-04259-f002:**
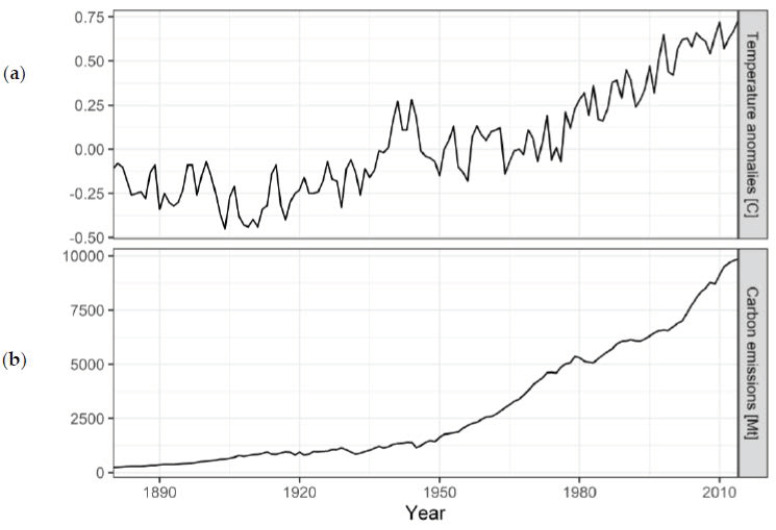
(**a**) Annual global temperature anomalies with respect to the 20th century mean and (**b**) annual global carbon emissions, 1880−2014.

**Figure 3 ijerph-18-04259-f003:**
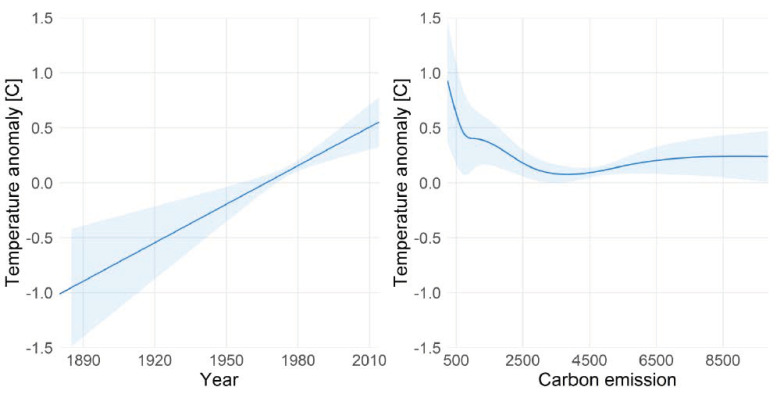
Partial effect plots of the independent variables year (**left**) and annual global CO_2_ emission (**right**). Shaded area illustrates the 95% confidence interval of the partial effect curve.

**Table 1 ijerph-18-04259-t001:** Illustrative extract of the data of the study of Ratzinger et al. [11]. White blood cells consist of the five subtypes neutrophils, eosinophils, basophils, lymphocytes, and monocytes and, hence, their sum equals the white blood cell count. See Appendix A for the full table and the analysis.

Observation	Neutrophils (G/L)	Eosinophils (G/L)	Basophils (G/L)	Lymphocytes (G/L)	Monocytes (G/L)	White Blood Cell Count (G/L)	C-Reactive Protein (mg/dL)
1	11.5	0.0	0.1	0.6	1.1	13.3	15.99
2	13.9	0.0	0.0	3.0	3.3	20.2	13.27
3	13.0	0.2	0.0	0.2	1.1	14.5	14.99
4	11.0	0.1	0.0	0.6	0.8	12.5	9.93
5	10.1	0.0	0.0	0.6	0.8	11.5	16.70

**Table 2 ijerph-18-04259-t002:** Odds ratios with 95% confidence intervals (CI), Akaike information criteria (AIC), and the C-statistics for the two fitted univariable logistic regression models and the multivariable model including both independent variables.

Model for Disease Severity	Independent Variable(s)	Odds Ratio		Model Performance	
		Estimate	95% CI	AIC	C-Index
Univariable model 1	GGO	1.82	(1.55, 2.22)	88.6	0.96
Univariable model 2	Consolidation	1.94	(1.59, 2.47)	142.8	0.89
Multivariable model	GGO Consolidation	1.83 0.99	(1.48, 2.38) (0.74, 1.33)	90.6	0.96


**Table 3 ijerph-18-04259-t003:** Some options to deal with collinearity by research aim. With ‘symptoms’, we mean typical consequences of collinearity such as inflated standard errors and unstable parameter estimates.

Method	Explanation	Remark
*Descriptive research aim*		
Variable omission	Omit one of the variables involved in the collinearity	Removes the symptoms, but leads to different interpretation of the model
Summary score	Combine several nearly collinear variables into a summary score and include only the summary score in the regression model	Removes the symptoms, retains most of the predictive value of the model, but leads to different interpretation of the model
*Predictive research aim*		
Use information criteria	Information criteria such as Akaike’s can be used to guide model building	Information criteria guide the analyst in a search for the most predictive model
*Explanatory research aim*		
Use causal reasoning	Specification of variables (exposure of interest, confounders) is necessitated by causal reasoning	Neither exposure nor confounders should be omitted as this violates assumptions needed to identify the causal estimand of interest

## Data Availability

The data and code presented in this study are openly available as teaching examples in the repository ‘CorrPred’ on Github (https://github.com/mgregorich/CorrPred, accessed on 15 April 2021).

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
