# Peer review of "Regression with Highly Correlated Predictors: Variable Omission Is Not the Solution"

_ijerph, 2021, doi:10.3390/ijerph18084259_

Round 1

Reviewer 1 Report

The article shows properly the problem of correlated data in lineal regression models. The authors present the statistical problem through cases of wide interest. My observations are the following:

  1. Line 86. Explain the term “I”
  2. Line 87. Explain the “R” superscript
  3. Equations 2 and 3. Check if it is correct to write the variable “y” in lowercase.
  4. Section 2.3. It is convenient to support with literature references what is expressed in lines 155 to 173.
  5. There must be a period written after (better).
  6. I am not clear about the methodology used in the research. It is convenient to place a diagram before that explains it.
  7. It is not clear if the investigation corresponds to a data analysis with case presentation. It is recommended being descriptive on this issue.
  8. It is advisable to expose clearly the contribution of the article to the state of art of statistics or the case of application.
  9. How do the authors consider the most recent knowledge about regression modeling theory in the presence of data that have the problem described in the article?
  10. Can the authors clarify which is the contribution that they pretend to expose? If it corresponds to a finding, method or discovery.

Author Response

Point-by-point Response to Reviewer comments

Reviewer 1

The article shows properly the problem of correlated data in lineal regression models. The authors present the statistical problem through cases of wide interest. My observations are the following:

Q: Line 86. Explain the term “I”

A: Thank you, done (line 94 in track-changes-visible version).

Q: Line 87. Explain the “R” superscript

A: Has been explained (line 94).

Q: Equations 2 and 3. Check if it is correct to write the variable “y” in lowercase.

A: Since here y corresponds to a vector of observations rather than to a matrix random variable, it is appropriate to write y in lowercase.

Q: Section 2.3. It is convenient to support with literature references what is expressed in lines 155 to 173.

A: We added appropriate references to this passage.

Q: There must be a period written after (better).

Q: We have rephrased the sentence in question. (line 237)

Q: I am not clear about the methodology used in the research. It is convenient to place a diagram before that explains it.

A: Since this paper is aiming at educating the readers and at providing guidance in statistical analysis of public health projects rather than to present a specific study, we were unsure how such a diagram could be designed. However, we find the reviewer comment very relevant and so we included a new Table 3 to support our message. (lines 418-419)

Q: It is not clear if the investigation corresponds to a data analysis with case presentation. It is recommended being descriptive on this issue.

A: Strictly following the topic of the special issue 'Statistical modelling: best practices for description, prediction and explanation' this paper aims at improving the practice of data analysis by evaluating variable omission as a tool to remove collinearity under different research aims. The data examples are chosen such that they best support our investigation.

Q: It is advisable to expose clearly the contribution of the article to the state of art of statistics or the case of application.

A: In the first paragraph of the discussion (lines 344-354) we now state what the contribution of this article is: By means of analysis of several examples, we demonstrate that it is the aim of the analysis that should guide decisions on keeping or omitting a variable in a model, and that such decisions should not be guided by common recommendations.

Q: How do the authors consider the most recent knowledge about regression modeling theory in the presence of data that have the problem described in the article?

A: We have added further articles that have dealt with collinearity in regression modelling from a more mathematical point of view (e.g. references 4 and 7). However, the problem has not yet been treated by a comparative analysis of studies with predictive and explanatory research aims. We have clarified this in the introduction. (lines 54-67)

Q: Can the authors clarify which is the contribution that they pretend to expose? If it corresponds to a finding, method or discovery.

A: Following the call of this special issue to provide guidance in statistical modelling, and according to how Reviewer 2 puts it, our contribution lies in 'demonstrating with two examples how the diagnostic tools for collinearity or near-collinearity may fail in guiding the analyst in selecting the most appropriate way of handling collinearity. We suggested that the multicollinearity handling should be driven by the research question at hand, in particular by the distinction between predictive or explanatory aims.' We clarified this by rephrasing the first paragraph of the discussion. (lines 344-350)

Reviewer 2 Report

The authors of submitted manuscript titled " Regression with highly correlated predictors: variable omission is not the solution" aimed to demonstrate with two examples how the diagnostic tools for  collinearity or near-collinearity may fail in guiding the analyst in selecting the most appropriate way of handling collinearity. The authors suggest that the multicollineraity handling  should be driven by the research question at hand, in particular by the distinction between predictive or explanatory aims. The content in most of the parts of the paper is scientifically well known.  The authors introduce collinearity and near-collinearity and refer to past studies conducted by Belsley  et al. (1980), Vatcheva et al. (2016), Graham M.H. (2003), Tu et al., (2004), Dormann et al.(2013), Leeuwenberg  et al. (submitted in 2021) and other  publications.

I have a few  main comments that should be addressed.

It seems that there are 3 empirical examples in this paper however the first is discussed in the introduction section. I suggest white blood cells counts example to be introduced in the Introduction section and further analysis and results to be presented in the results section.

In the first example the authors present a Covid-19 study where statistical modeling is used to pursue a predictive aim in the case of  two highly correlated predictors included in the model. As a result the authors demonstrate how AIC may be useful to guide model building, but collinearity checks are not sufficient to make decisions on variable selection.   However, the VIF rules of thumb should be interpreted with cautions (Vatcheva et al., 2016; O’Brien, 2007).

Other formal multicollinearity diagnostic measures that can be used to evaluate for potential multicollinearity effect, not evaluated in this analysis, are the condition index assisted by the regression coefficients variance-decomposition proportion.  High variance decomposition-proportion of two or more regression coefficients associated with a high condition index indicates which variables are potentially involved in the multicollinearity.

In addition, as shown in Table 2, variable Consolidation  is no longer significantly associated with the outcome in the Multivariable model (OR=0.99, 95% CI: 0.74, 1.33). The change in the estimated regression coefficients for variable Consolidation after adjusting for variable GGO is approximately -102%. This could be a sign of potential  multicollinearity as well. What are the corresponding p-values for the models in Table 2?

Where are the results from  the redundancy analyses? Proper references  should be added to the respective tables or graphs.

In example #2, the authors discuss a valid point that in some cases the omission of one of the variables involved in the multicollinearity effect will not be a possible solution in order to answer the study question, also discussed by O'Brien (2017) cited by the authors. As formal diagnostics of multicollinearity the condition index assisted by the regression coefficients variance-decomposition proportion can be presented as well. Where are the numerical results from the generalized additive model? 

Author Response

Point-by-point Response to Reviewer comments

Reviewer 2

Q: The authors of submitted manuscript titled " Regression with highly correlated predictors: variable omission is not the solution" aimed to demonstrate with two examples how the diagnostic tools for  collinearity or near-collinearity may fail in guiding the analyst in selecting the most appropriate way of handling collinearity. The authors suggest that the multicollineraity handling  should be driven by the research question at hand, in particular by the distinction between predictive or explanatory aims. The content in most of the parts of the paper is scientifically well known.  The authors introduce collinearity and near-collinearity and refer to past studies conducted by Belsley  et al. (1980), Vatcheva et al. (2016), Graham M.H. (2003), Tu et al., (2004), Dormann et al.(2013), Leeuwenberg  et al. (submitted in 2021) and other  publications.

A: We thank the reviewer for nicely summarizing our aims and we took the liberty to partly use this summary in order to better clarify our contribution in the first paragraph of the discussion. (lines 344-350 in the track-changes-visible version)

Q: I have a few  main comments that should be addressed. It seems that there are 3 empirical examples in this paper however the first is discussed in the introduction section. I suggest white blood cells counts example to be introduced in the Introduction section and further analysis and results to be presented in the results section.

A: We use the blood analysis example as a 'toy example' to explain the issues of exact collinearity and near-collinearity. This example serves well to more informally explain these concepts and their differences that are described in the Methods section, which is essential in this journal as many readers may not be familiar with statistical theory. Therefore, the example will also be mentioned in the Methods part. The supplement contains a fully reproducible R markdown report for the interested reader. Moreover, in the paper we provide the link to a GitHub repository that contains all data sets and R scripts to reproduce our results. (see 430-436, 437-438)

Q: In the first example the authors present a Covid-19 study where statistical modeling is used to pursue a predictive aim in the case of  two highly correlated predictors included in the model. As a result the authors demonstrate how AIC may be useful to guide model building, but collinearity checks are not sufficient to make decisions on variable selection.   However, the VIF rules of thumb should be interpreted with cautions (Vatcheva et al., 2016; O’Brien, 2007).

Other formal multicollinearity diagnostic measures that can be used to evaluate for potential multicollinearity effect, not evaluated in this analysis, are the condition index assisted by the regression coefficients variance-decomposition proportion.  High variance decomposition-proportion of two or more regression coefficients associated with a high condition index indicates which variables are potentially involved in the multicollinearity.

A: We now computed and report the condition number in all examples and discuss its use. We appreciate that condition indices supplemented with regression coefficient variance-decomposition proportions provide a useful tool that may give even more insight in multivariable situations where the number of variables is greater than two. To keep the presentation simple, we chose to focus here on the condition number only. In the R markdown report for the blood analysis example, also an analysis with the suggested tools has been included. (see the new paragraph lines 154-163)

Q: In addition, as shown in Table 2, variable Consolidation  is no longer significantly associated with the outcome in the Multivariable model (OR=0.99, 95% CI: 0.74, 1.33). The change in the estimated regression coefficients for variable Consolidation after adjusting for variable GGO is approximately -102%. This could be a sign of potential  multicollinearity as well. What are the corresponding p-values for the models in Table 2?

A: We now explicitly mention that the change in estimated regression coefficient of roughly -100% is a sign of potential multicollinearity as well. (lines 232-236)  We abstain from presenting p-values in Table 2 as it is commonly accepted that they are irrelevant for prediction when no specific hypothesis is tested. Moreover, they do not provide information in addition to the confidence intervals.

Q: Where are the results from  the redundancy analyses? Proper references  should be added to the respective tables or graphs.

A: We now report the results from redundancy analysis. Redundancy analysis is explained in Frank Harrell's book 'Regression Modeling Strategies', 2nd edition, on page 80, and we assume it is quite popular as it has been implemented in a function redun included in Frank Harrell's famous Hmisc package. The output of that function consists of the R-squares with which each variable can be predicted using generalized additive models with all other independent variables (in our examples, there is always only one other independent variable). The R-square values for predicting both variables with the respective other variable are now explicitly stated. We also give the proper reference. In Supplemental Material 2, the R Markdown report for the blood analysis example, we demonstrate application of redun. (see 176-178, 227-229)

Q: In example #2, the authors discuss a valid point that in some cases the omission of one of the variables involved in the multicollinearity effect will not be a possible solution in order to answer the study question, also discussed by O'Brien (2017) cited by the authors. As formal diagnostics of multicollinearity the condition index assisted by the regression coefficients variance-decomposition proportion can be presented as well. Where are the numerical results from the generalized additive model? 

A: Since we only have two variables in each of the examples, we included the condition number as a sufficient summary and referred the interested reader to the book of Belsley et al (reference 1). (see lines 154-163) However, it does not change our main thesis that it is the aim of the analysis, not the type of collinearity diagnostic, that should guide data analysis. Numerical results from the worked example in section 3.2 have been included in Supplemental Material 1. In particular, we state the regression coefficients and their standard errors that were estimated and which underlie the partial effect plots of Figure 3. (Supplemental Material 1)

Round 2

Reviewer 1 Report

The content is interesting. For a general audience, it would be satisfactory. This paper can be considered as a research note.

Author Response

We thank the reviewer for again assessing our manuscript. In response to tje suggestion to improve some parts of the manuscript, we have again proofread the manuscript and made some cosmetic changes to make the presentation clearer (see tracked changes).